# Infrared Spectroscopy of Urine for the Non-Invasive Detection of Endometrial Cancer

**DOI:** 10.3390/cancers14205015

**Published:** 2022-10-13

**Authors:** Carlos A. Meza Ramirez, Helen Stringfellow, Raj Naik, Emma J. Crosbie, Maria Paraskevaidi, Ihtesham U. Rehman, Pierre Martin-Hirsch

**Affiliations:** 1Bioengineering, School of Engineering, Faculty of Science and Technology, Lancaster University, Gillow Avenue, Lancaster LA1 4YW, UK; 2Gynaecological Oncology, Clinical Research Facility, Lancashire Teaching Hospitals, Sharoe Green Lane, Preston PR2 9HT, UK; 3Gynaecological Oncology Research Group, Division of Cancer Sciences, University of Manchester, Oxford Road, Manchester M13 9WL, UK; 4Department of Obstetrics and Gynaecology, St Mary’s Hospital, Manchester University NHS Foundation Trust, Manchester Academic Health Science Centre, Oxford Road, Manchester M13 9WL, UK; 5Reproductive and Developmental Biology (IRDB), Department of Metabolism, Digestion & Reproduction, Imperial College London, London W12 0HS, UK

**Keywords:** machine learning, endometrial cancer detection, PLS-DA, cancer, vibrational spectroscopy, ATR, FTIR

## Abstract

**Simple Summary:**

The incidence of endometrial cancer has increased across the western world, and with it the need for fast and efficient diagnostic methods. This study explores the potential of implementing a non-invasive test for the early detection of endometrial cancer. The concept of this new diagnostic alternative involves the analysis of carefully collected urine samples from female patients, together with infrared spectroscopy and statistical methods that allow the identification and prediction of the samples. Interestingly, we were able to obtain some of the spectral biomarkers that may be useful in the detection of endometrial cancer, such as spectral bands of DNA, or phenyl group vibrations that have been found to be highly linked to cancer. Additionally, the predictive values rise to over 90% specificity and sensitivity, making this technique an excellent technique to be explored in clinical studies, now that biomarkers have been identified in uncontaminated urine. These promising results in predicting and discriminating classes of cancer, suggest that this technique should be evaluated in pragmatic studies in cancer triage clinics.

**Abstract:**

Current triage for women with post-menopausal bleeding (PMB) to diagnose endometrial cancer rely on specialist referral for intimate tests to sequentially image, visualise and sample the endometrium. A point-of-care non-invasive triage tool with an instant readout could provide immediate reassurance for low-risk symptomatic women, whilst fast-tracking high-risk women for urgent intrauterine investigations. This study assessed the potential for infrared (IR) spectroscopy and attenuated total reflection (ATR) technology coupled with chemometric analysis of the resulting spectra for endometrial cancer detection in urine samples. Standardised urine collection and processing protocols were developed to ensure spectroscopic differences between cases and controls reflected cancer status. Urine spectroscopy distinguished endometrial cancer (*n* = 109) from benign gynaecological conditions (*n* = 110) with a sensitivity of 98% and specificity of 97%. If confirmed in subsequent low prevalence studies embedded in PMB clinics, this novel endometrial cancer detection tool could transform clinical practice by accurately selecting women with malignant pathology for urgent diagnostic work up whilst safely reassuring those without.

## 1. Introduction

Endometrial cancer is the sixth most common cancer in women worldwide, with more than 380,000 new cases diagnosed in 2018. Incidence rates are climbing, with the UK reporting a 60% increase in diagnoses over the last 30 years [1,2]. The main aetiological drivers are advanced age and obesity, with global shifts in the prevalence of these risk factors explaining recent epidemiological trends [3]. Although most women present with early stage disease, approximately 10–15% have advanced disease at diagnosis, and current mortality rates are projected to rise by 20% within the next 15 years [4].

Most endometrial cancers present with postmenopausal bleeding, a red flag symptom that triggers urgent specialist referral for a series of intimate tests that image, visualise and sample the endometrium [5]. These tests are expensive, time consuming and cause considerable anxiety [6]. Transvaginal ultrasound has excellent sensitivity but poor specificity (up to 50%), exposing a large proportion of symptomatic women to additional tests [7]. Outpatient hysteroscopy and endometrial biopsy are invasive procedures that are variably tolerated and commonly fail due to technical issues, high pain scores or non-diagnostic results, mandating repeat tests, sometimes under general anaesthesia, and prolonging timelines [8]. Since just 5–10% of those investigated for postmenopausal bleeding having sinister underlying pathology, better selection of patients for testing would streamline diagnostic services, improve patient experience and transform clinical care.

Vibrational spectroscopy is a powerful bio-analytical tool that shows great promise for detecting cancer [9]. The vibrations caused in molecular bonds from different energy sources are recorded as spectral ‘fingerprints’, which are characteristic for each sample composition [10]. Attenuated Total Reflection-Fourier Transform Infrared (ATR-FTIR) is a frequently used spectroscopic technique that produces different absorbance intensities in several wavelengths of the electromagnetic spectrum following molecular excitation from an infrared source [11]. Its application in easily collectable biofluids (particularly blood and urine) has shown considerable potential for detecting several types of cancer [12,13,14,15,16], including endometrial cancer [17,18,19]. These biofluids are ideal for cancer detection through spectroscopy, as their acquisition is minimally or non-invasive with negligible cost [20].

In a systematic review of clinical diagnostic studies using vibrational spectroscopy (including 94 cancer studies), only four interrogated urine as a potential biofluid for detecting cancer [21]. These four studies demonstrated high levels of diagnostic accuracy in head & neck cancer and oesophageal cancer [14,22,23,24,25]; a pilot study of urine spectroscopy for ovarian and endometrial cancer detection, which showed promising results, was excluded from this review due to small numbers [18]. Here, we aimed to determine the potential for ATR-FTIR spectroscopy to distinguish endometrial cancer patients from benign controls using urine samples in a large case–control study. The study aims to see if there are unique biomarkers in uncontaminated urine before pragmatic clinical studies.

## 2. Methods

### 2.1. Study Design

Participants were recruited from the Benign and Tertiary Oncology Services within the Gynaecology Department at Lancashire Teaching Hospitals between April 2018 and October 2019. Ethical approval for sample acquisition was granted by the East of England-Cambridge Central Research Ethics Committee (ref: 16/EE/0010; IRAS project ID: 195311) and for experimental analysis by Lancaster University (ref: STEMH 1073). Patients were eligible to take part if they were undergoing gynaecological surgery for any benign indication (controls) or for treatment of biopsy-proven endometrial cancer (cases). All participants provided written, informed consent to take part. Participants were assigned a unique study ID to retain their confidentiality.

### 2.2. Sample Acquisition, Processing and Storage

Urine specimens were collected immediately after patients were anaesthetised for surgery using a standardised protocol. All participants had fasted for at least six hours prior to urine collection. The external urethra was cleaned with a swab soaked with sterile water and the urine sample collected in a sterile dry pot using a disposable ‘in/out’ catheter. Once the urine sample had been collected, the patient was prepped for surgery in the normal way. Urine samples were frozen within two hours of collection and stored at −80 °C pending analysis. Prior to spectroscopic analysis, samples were allowed to thaw to room temperature. Once completely thawed, 60μL of urine was pipetted onto sterile FisherBrand^TM^ glass slides and dried at room temperature, as previously described [11,20].

### 2.3. Histopathological Assessment

Surgical specimens were formalin fixed and paraffin embedded for histological examination. All benign specimens were examined by at least one consultant gynaecological pathologist and malignant specimens at least two, followed by discussion at the gynaecological oncology multidisciplinary team meeting. Molecular profiling of endometrial cancers was only introduced in the last 24 months after the collection of the majority of these specimens and therefore tumours were not classified to molecular profile. We also combined high grade endometrial endometroid cancers with lesser grades of endometroid disease when comparing endometroid disease to other histological types. Staging of cancers was performed according to the International Federation of Gynaecology and Obstetrics (FIGO) systems for endometrial carcinomas [26] and uterine sarcomas [27]. Endometrial cancer was staged according to the International Federation of Gynaecology and Obstetrics (FIGO) staging system.

### 2.4. Spectral Acquisition

ATR-FTIR spectra were obtained with a Bruker TENSOR 27 FTIR spectrometer and a Helios ATR attachment, equipped with a diamond crystal (Bruker Optics Ltd., Coventry, UK) and operated using OPUS 6.5™ software. Spectra for each sample were acquired from 10 different points at 8 cm^−1^ resolution and accumulating 32 scans for each measured point. The ATR diamond crystal was cleaned with distilled water and dried with tissue paper between samples. A background spectrum with the same parameters described above was taken prior to each new sample analysis.

### 2.5. Computational Analysis

Prediction analysis was performed in Python v.3.7.4™, in conjunction with scikit-learn v1.0. kit-learn library, and pypols v. 20.3.post1. Second polynomial order with 20 iterations was used for baseline correction. Acquired spectral data was pre-processed prior to the cluster analysis and label prediction. For statistical and multivariable analysis, spectra were defined and truncated at the fingerprint region (1800–500 cm^−1^).

Spectra were normalised by standard normal variate (SNV) before application of the clustering analysis, principal component analysis (PCA) and PCA-K means. PCA score plots were generated as an initial exploratory analysis to evaluate the effect of potential confounding factors, such as age (<60 years; ≥60 years), BMI (<30 kg/m^2^; ≥30 kg/m^2^) and type 2 diabetes mellitus status (diabetes; no diabetes). The number of clusters in the PCA-K-means model was determined by the ‘elbow point’ method, the optimal number of clusters where the WCSS (Within-Cluster Sum of Square) starts decreasing in a linear fashion (Appendix A) [28].

Orthogonal projection of latent structures (OPLS) was employed as a pre-processing technique for the supervised analysis; 48 orthogonal components were implemented based on the parameters selected by BirG (2020) and Worley B. et al. (2016) [29,30]. Spectra were split into 70% training set and 30% test set to be able to make prediction with the supervised models. Partial least squares discriminant analysis (PLS-DA) was applied to the OPLS transformed spectra. PLS regression coefficients were obtained for each class comparison. PLS regression coefficients is a useful tool to evaluate the influence of the variables in the model, coefficient scores which are greater than zero (positive or negative) contribute to the discrimination, the higher the score, the higher the contribution to the discriminatory analysis [31]. Herein, the three coefficients with the highest and lowest scores assigned to the spectral features were chosen as the peaks that were mostly responsible for the observed discrimination between the different groups. Since, PLSA-DA can be implemented to perform prediction tests, and given its friendly behaviour with low amounts of data, the highest and lowest regression coefficients were used for prediction of classes, considering a 10 fold cross-validation for the PLS-DA model [32].

### 2.6. Statistical Analysis

Statistical analysis of spectral features was performed using IBM SPSS 27 software with all tests performed on the pre-processed spectra within the fingerprint region considering significant a *p*-value < 0.05. Assessment of normality and multi-collinearity of the data was performed using the Shapiro–Wilk test and Pearson correlation, respectively. More specifically, data was considered to follow a normal distribution when *p* values > 0.05, while multi-collinearity was assumed when the Pearson correlation coefficient was greater than 0.9. A multivariate analysis of variance (MANOVA) test was applied for the existing confounders (i.e., age, BMI, diabetes status) whereas the statistical significance of the discriminatory spectral features was determined by a t-test or Mann–Whitney U test, depending on whether the data followed a normal or non-normal distribution.

Prediction metrics (sensitivity, specificity, positive predictive value and F1 Score) were obtained by scikit-learn python library ‘classification report’, and the acquisition of these metrics is based on the formulas illustrated at equations in Appendix A. On the other hand, accuracy of prediction was calculated using scikit-learn python library ‘accuracy score’.

## 3. Results

The study population comprised 109 women with endometrial cancer (cases) and 110 with benign gynaecological conditions (Table 1). The cases were significantly older than the controls (median age 67 vs. 56 years, *p* < 0.05), and more likely to have obesity (median BMI 31.9 vs. 27.8 kg/m^2^, *p* < 0.05) and type 2 diabetes mellitus (19 vs. 9, *p* < 0.05) (Table 2). Most endometrial cancers were stage I (68.8%) low grade (44.9% grade 1 or 2) endometrioid (52.2%) tumours, in keeping with national statistics, although there was a relatively high proportion of non-endometrioid and higher stage disease due to Lancashire Teaching Hospitals status as a regional gynaecological cancer centre. The controls had a range of benign gynaecological conditions, including fibroids, endometriosis and ovarian cysts, or no absence of any pathology (*n* = 15, 13.6%).

### 3.1. Endometrial Cancer Detection

Spectra were collected from urine samples and the data were interrogated for patterns that could distinguish cases from controls. The advantage of utilising PCA as a preprocessing step for k-means analysis, is that the dimensions of the input variables from the spectra, reducing the variable noise [33]. The reduced dimensions from the data provides, helps to improve the cluster selection of the k-means clustering method which allows an initial overview of the differentiation of the ATR-FTIR spectra obtained in this study. According to the selection of the best number of clusters (k = 4), illustrated in Appendix A, it can be observed in Figure 1 that four clusters can be identified in the data. The importance of evaluating the data with a clustering method, such as k-means, lies in the fact that it allows to identify the presence of different groups within the data [32,33]. However, due to the nature of the clustering model, it is not possible to know to which class each cluster belongs. Nonetheless, this tool allows us to assess whether the analysed data may have discriminating patterns [33,34].

Moreover, four clusters could be differentiated (Figure 1), potentially, (i) controls, (ii) endometrioid cancers, (iii) non-endometrioid cancers and (iv) a sub-differentiation within the cancer types. The clusters on their own are not radically apart from each other, however, comparing the result obtained in Appendix A with the gap statistic in Appendix A, will allow to provide a meaningful and reliable result on the k value selection. Gap statistic measures the cluster performance based on the mean dispersion compared to a reference with uniform distribution for an increasing number of clusters [28].

Differently from PCA-K means cluster analysis, discrimination plots were obtained by PLS-DA, these discriminant analysis plots compared the binary distribution of the classes of interest. Figure 2 shows the trained samples used for prediction. All class comparisons; controls vs. all cancers (Figure 2A), controls vs. endometrioid cancers (EC) (Figure 2B), controls vs. non-endometroid cancers (NEC) (Figure 2C), endometrioid cancers vs. non-endometroid cancers (Figure 2D), controls vs. stage 1 cancers (Figure 2E), and controls vs. stage 1A grade 1 cancers (Figure 2F) show major discrimination. Interestingly, for all comparisons *p* < 0.0001, indicating significant difference between classes.

Alternatively, receiver operating characteristic (ROC) curves and their corresponding area under the curve (AUC) were obtained (Figure 3) to evaluate not only the relationship between the true positive ratio and false positive ratio (FPR), alternatively named sensitivity and 1-specificity, but also the accuracy of the discrimination analysis [32,35].

Furthermore, the AUC between controls and all cancer groups(Figure 3A) presented an outstanding discrimination of 99.32%, with a sensitivity of 95.29% and a specificity of 98.89% (FPR = 0.0111). On the other hand, for the groups; controls vs. endometrioid cancers, endometrioid cancers vs. non-endometrioid cancers, and controls vs. stage 1A grade 1 cancers (Figure 3B,D,F) the AUC, sensitivity, and specificity were 100%. In contrast, the ROC/AUC of the discrimination between controls vs. non-endometrioid cancers had a sensitivity of 100% and a specificity of 98.71% (FPR = 0.01282) and an AUC of 99.96% (Figure 3C). The comparison between control vs. stage I cancers showed a sensitivity of 98.11%, a specificity of 97.36% (FPR = 0.02631) and an AUC of 99.73% (Figure 3E). Overall, the AUC in conjunction with the cluster discrimination analysis performed by PLS-DA of the latter classes, demonstrated that these methods can efficiently discriminate clusters between classes.

Additionally, PLS-DA was also used to perform prediction analysis. Appendix A in the ESI suggests, the lowest prediction accuracy was found when predicting labels between endometroid cancers vs. non-endometroid cancers (71.63% SD +/− 18.29%), whereas the highest (100% SD +/− 0%) was observed between controls vs. stage IA grade 1 cancers. Moreover, the remaining binary predictions present outstanding predictions as illustrated in Appendix A. The predictions performed for the controls vs. endometroid cancers, and controls vs. NEC groups possess outstanding metrics, since the sensitivities and specificities range between 86–100%, and 90–100%, respectively. Furthermore, although the accuracy prediction for the binary dataset between controls vs. all cancers have a lower performance in comparison with the rest of the aforementioned classes, the accuracy of prediction outcome illustrated in Appendix A indicates a promising performance.

### 3.2. Prospective Spectral Biomarkers

As it was previously described in the methods section, the most discriminative spectral bands were identified from the regression coefficients after PLS-DA (Appendix A). To provide a visual comparison from the identified spectral bands, the absorbance intensities were reported in Figure 4. The complete table of tentative spectral assignments for identification of biomarkers can be found on Appendix A.

The *p*-value for all discriminated spectral bands was *p* < 0.0001. Moreover, the statistical significance between the medians of each class of each spectral band is demonstrated by the SD of each bar that represents the absorbance. The mean of the absorbance which was obtained for each class, showed an appropriate comparison between the identified biomarks.

A noteworthy attribute of the molecular markers identified by infrared spectroscopy, illustrated in Figure 4, is that the absorbance of the cancer markers is higher than that of the control. Interestingly, in the comparison made between endometrioid cancer vs. non-endometrioid cancer, endometrial cancer has a higher absorbance of molecular spectral markers compared to non-endometrial cancer. From the data illustrated in Figure 4 it can be seen that the assignment of lipids between 1762 and 1797 cm^−1^ (C=O, C=C stretching [36,37]) is a common marker in all the comparisons made. Additionally, protein markers found in the comparisons of all cancers vs. controls (Figure 4A) (Amide I, 1662 cm^−1^ [36]), endometrioid vs. non-endometrioid (Figure 4D) (Amide I, 1627 cm^−1^ [36]), endometrioid vs. controls (Figure 4B) (CH_3_ asymmetric bending of proteins, 1450 cm^−1^ [36,37]), stage IA grade 1 cancers vs. controls (Figure 4F) (Amide III, 1346 cm^−1^ [36,37]), and non-endometrioid vs. controls (Figure 4C) (CH_2_ wagging of collagen, 1338 cm^−1^ [36]) can be observed. Furthermore, DNA bands are found on all class comparisons, except cancers vs. controls, and endometrioid vs. non-endometrioid cancers, between 825–1114 cm^−1^. Although there are different biomolecule markers that can be prospective for differentiating between cancer types vs. controls, it can be observed that the highest number of bands identified in the classes are attributed to phenyl ring vibrations, and CH out of plane vibrations, and deformations attributed to aromatic molecules [36,37]. These bands are found on 1585 cm^−1^ (Phenyl ring deformation [36]), 1485 cm^−1^ (CH deformation [36]), 810 cm^−1^ (Ring CH deformation [36,37]), 786 cm^−1^ (CH out of plane bending vibration [36,37]), 763 cm^−1^ (CH out of plane bending vibration [36,37]), 686 cm^−1^ (CH out of plane bending vibration [36,37]), 675 cm^−1^ (CH out of plane bending vibration [36,37]), 628 cm^−1^ (CH out of plane bending vibration [36,37]), 609 cm^−1^ (Ring deformation of phenyl [36,37]), and 520 cm^−1^ (Cα = Cα’ torsion and ring torsion of phenyl [36,37]).

Interestingly, the difference in absorbance between the prospective biomarks found for control vs. all cancers, and EC vs. NEC (Figure 4A,D) was found to be minimal. Since the *p*-value for these bands was *p* < 0.000, what is important to highlight is that these groups have in common the presence of phenyl vibrations, and amide vibrations (Appendix A).

The identification of these prospective biomarkers is illustrated in Figure 5. This figure, shows how the absorbance of the cancer groups in the spectrograms differs in intensity from the control group. The OPLS model, in conjunction with PLS regression, makes it possible to identify which spectral bands have the greatest influence on the segregation of the data. Interestingly, the prospective spectral biomarks are found close to each other (shaded regions in Figure 5), indicating that not only the prospective biomarks are a good indicator of endometrial cancer, but also that these regions possess a huge potential for future endometrial cancer diagnostic methods.

## 4. Discussion

Most women with endometrial cancer present with post-menopausal bleeding (PMB), however only 5% of women with PMB have underlying malignant pathology. There are currently 9700 women diagnosed with endometrial cancer in the UK each year and investigating women with PMB promptly challenges cancer diagnostic resources. Women with PMB are currently triaged in the UK by having an initial clinical examination and then a transvaginal scan assessing the thickness of endometrium, and if the thickness of the endometrium is suspicious, a subsequent out-patient hysteroscopy and endometrial biopsy are performed. This pathway has its limitations; transvaginal scan lacks specificity as a triage tool exposing a high proportion of women who do not have endometrial cancer to further tests. Failed out-patient hysteroscopy and endometrial biopsy, mostly due to intolerable pain or technical failure, affects up to 31% of women, requiring an additional procedure under general anaesthetic. Many women avoid presentation due to the invasive and intimate nature of transvaginal scan and out-patient hysteroscopy. Some ethnic minority elderly women are particularly apprehensive about such examinations and delay their presentation to health care. Currently, thousands of women who do not have endometrial cancer undergo these invasive procedures every year, with significant financial implications to the health economy and considerable personal cost to women. The large volume of women with post-menopausal bleeding often challenges resources and often impacts on prompt diagnosis. Therefore, a robust objective non-invasive inexpensive urine test offers the potential of a rapid triage test with high diagnostic accuracy and has the potential of speeding up cancer diagnosis (acceptable to more women) whilst at the same time re-assuring women without the disease that they are well.

Our results of this pilot study have demonstrated exceptionally high levels of diagnostic accuracy in differentiating between urine from women with endometrial cancer of all histological sub-types and benign controls. The results were consistent when comparing the most common endometrioid cancer to controls, and non-endometrioid (serous, clear cell and carcinosarcoma) cancers to controls. Furthermore, when we compared early stage I cancers to control and grade 1 endometrioid stage IA cancers to controls (earliest stage and grade of disease) there was still very high levels of diagnostic accuracy. Clearly, in the latter comparison this was a small sub-set of all the cancers and might be statistically less powerful, but the diagnostic accuracy remained exceptionally high.

The use of OPLS as pre-processing method, coupled with PLS-DA, has proven to provide a useful way to discriminate classes from the collected spectra allowing outstanding specificities and sensitivities, above 90%, and 83%, respectively. This method is commonly and successfully used in the biological sector in conjunction with PLS, PLS-derived methods, as well as other machine learning methods [38,39,40]. In contrast with PCA-K means clustering method, the PLS-DA plots evaluate a binary supervised segregation between the studied classes. Both are different clustering methods with different approaches, however PLS-DA is able to distinguish trained data variation between classes.

Additionally, as observed in Figure 4, the most discriminatory peaks mean absorbances are found to be higher in all cancer groups in comparison with the control groups, this feature provides a great insight on the impact on the molecular analysis, since high absorbances on the cancer groups suggests an increased metabolic activity [41]. In contrast, the comparison between endometrioid vs. non-endometrioid cancers have a similar behaviour, where endometrioid cancers show a higher absorbance in the most influential bands. Although the absorbances between each spectral bands are not huge, the obtained *p* value of *p* < 0.0001 suggests a statistically significant difference between these groups and as previously noted, the significant change in the absorbance could be correlated to the differences between molecules processed by the metabolic activity of the assigned cancer [41]. Although there are molecular differences between endometrioid and non-endometrioid cancers [42], further studies in bio-spectroscopy should be performed in these groups using urine samples.

Prospective biomarkers for classification between cancer cases and controls included all major biomolecules (proteins, carbohydrates, lipids, nucleic acids, Appendix A). Urinary peptides and glycoproteins have been suggested as potential biomarkers for identification of endometrial cancer, among which urinary epidermal growth factor (EGF) has been found to be up-regulated in endometrial cancer patients [43,44]. The advantage of peptides excreted in urine lies in their reduced complexity and higher stability compared to blood plasma proteins, rendering them ideal substrates for biomarker identification [45]. Increased concentrations of urinary lipids have been identified in renal and prostate cancer patients, and these effects may be present in endometrial cancer as well [46,47]. Increased levels of urinary nucleic acids may reflect the unconstrained proliferation of cells in malignant states, and significantly higher expression (up to 30 fold) has been identified in urinary exosomes isolated from endometrial cancer patients compared to controls [48]. Furthermore, one unanticipated finding was that several prospective bands for endometrial cancer classification, were found at 1485 cm^−1^ and within 810–520 cm^−1^, assigned and linked to phenyl groups. There are two key findings that must be considered to explain the presence of these vibrations in urine samples. Firstly, phenylalanine as essential amino acid happens to be metabolised and excreted by the urinary tract and therefore able to detect in this biofluid [49]. It has been demonstrated that there is a significant relationship between high levels of phenylalanine due to a deficiency of phenylalanine hydroxylase and endometrial cancer, since it could affect the development or progression of endometrial cancer [50]. Secondly, it has been found that naturally and synthetically occurring aromatic groups, are related to antitumoral, antimetastatic, and antiangiogenic activities [51,52].

We tested to see if high BMI, diabetes and age influenced the results as all these can pre-dispose to the development of endometrial cancer, none of these factors influenced the results (Appendix A).

In contrast to other studies that have characterised endometrial cells being exfoliated into the vagina and consequently shed into vaginal secretions or contaminate urine [53], this study has been performed by acquiring urine specimens at the time of surgery for either endometrial cancer or benign conditions. Collection of urine specimens was achieved by cleaning the external urethra with a swab soaked with sterile water (before cleaning the external genitalia and vagina with iodine) and inserting a sterile tube into the bladder to obtain the urine specimen. Thus, the specimen was not contaminated with any secretions or vaginal debris, as urine was collected directly from the urinary bladder. Therefore, the spectra acquired were from neat urine alone. We did not send urine specimens to exclude urinary tract infections prior to surgery as this was a pragmatic study, and it is not routine clinical practice to do so. Demonstration of clear differences between controls and malignant cases, and other comparisons is therefore reliant on biomarkers excreted in urine and not contamination by exfoliated cells or any other debris. Studies evaluating cell characterisation of urine or vaginal fluid have looked at the morphology of cells, which is subjective or dependent on cellular debris either in urine or on collection devices. These studies have also been conducted using the ‘first’ early morning urines to ensure maximum quantities of exfoliated endometrial cells in urine and vaginal fluid. In contrast, in our study of spectroscopy on neat urine, the biomarkers are independent of debris and results were consistent regardless of the timing of surgery, which was undertaken either in the morning or in the afternoon. There was no restriction on patients passing urine before surgery.

## 5. Conclusions

This pioneering study represents a potential breakthrough in the early diagnosis of endometrial cancer. A robust objective non-invasive test that has very high diagnostic accuracy has the potential to be an effective triage test for women with post-menopausal bleeding or indeed a screening test for asymptomatic women at risk of endometrial cancer. In England, the NHS 2021 Cancer strategy aims to improve cancer outcomes by detecting cancers early. To that aim, the NHS strategy ambition is to develop community diagnostic centres to facilitate earlier diagnosis of disease. Urine testing is non-invasive and not intimate facilitating a greater uptake in all women regardless of age or religious beliefs and would be ideal as a rapid triage test within this Cancer strategy. These early results will need to be reproducible in definitive large pragmatic diagnostic studies across multiple post-menopausal bleeding clinics that are designed to fast track symptomatic women. Women with PMB have a 5–10% rate of endometrial cancer and the results from this current proof of principal study need to be verified in a low cancer prevalence setting. Modern technology has facilitated the miniaturisation of FTIR equipment and processing of spectra, and analysis can be done within minutes in conjunction with any prediction model such as PLS-DA. This technology can therefore give a real time result in either a primary care or secondary setting. Clearly, future studies should evaluate if a mid-stream specimen of urine taken in a clinic setting gives similar results to those taken at the time of surgery (or whether the first urine in the morning is equally robust). Further studies should also evaluate, if spectroscopic techniques can differentiate different molecular profiles of endometrial cancers and whether biofluid detection is influenced by such categorisations.

## Figures and Tables

**Figure 1 cancers-14-05015-f001:**
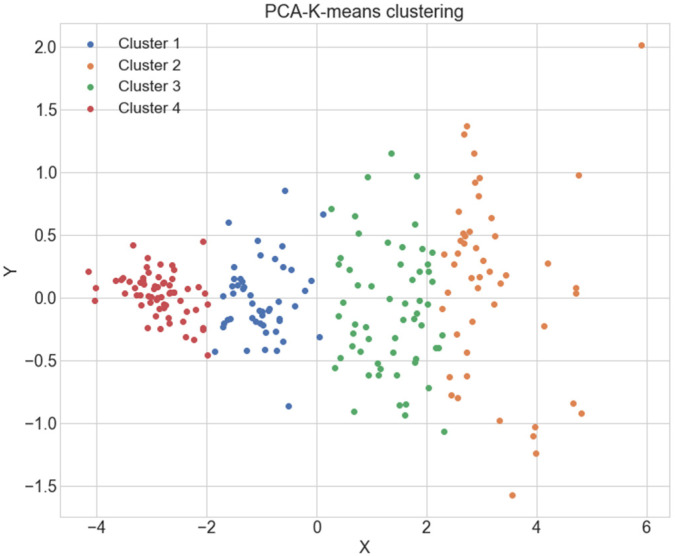
PCA-K-means of the spectra within the fingerprint region. Four clusters are identified by using the elbow method and corroborated the gap statistic. The presence of clusters allows to determine whether the data actually have any possible segregated data.

**Figure 2 cancers-14-05015-f002:**
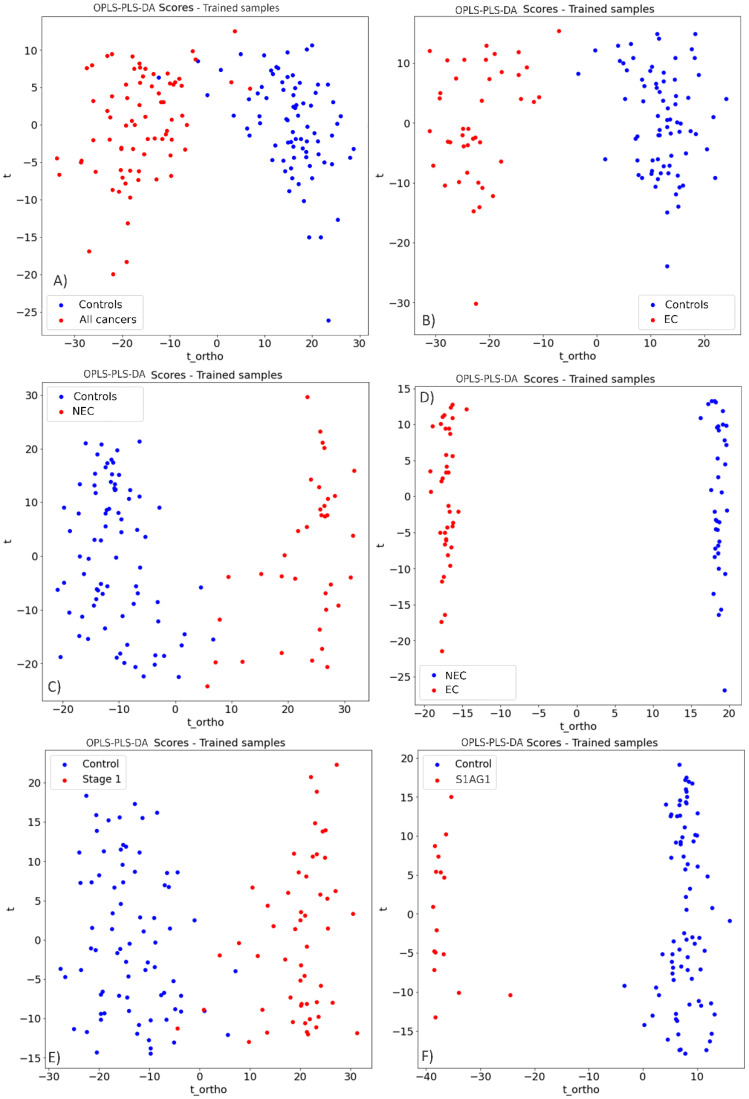
Discriminant analysis plots of all class comparisons. (**A**) Controls vs. all cancers. (**B**) Controls vs. endometrioid cancers (EC). (**C**) Controls vs. non-endometrioid cancers (NEC). (**D**) Endometrioid cancers vs. non-endometrioid cancers. (**E**) Controls vs. stage I cancers. (**F**) Controls vs. stage IA grade 1 cancers. After the application of PLS-DA it was possible to discern between classes, obtaining a *p* < 0.0001 shows statistical significance on the discrimination analysis.

**Figure 3 cancers-14-05015-f003:**
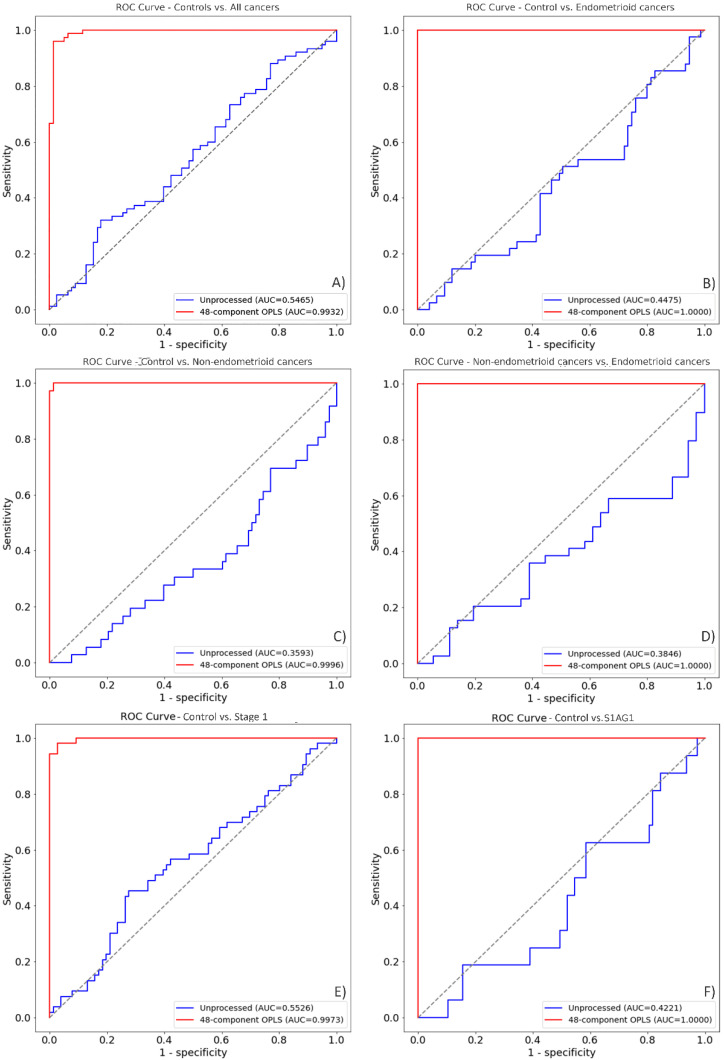
Receiver operating characteristic (ROC) curves after the application of OPLS-PLS-DA. Figures (**A**–**F**) illustrate the ROC curve of OPLS-PLS-DA processed data (red curves) and unprocessed data (blue curves). These figures demonstrate how the discrimination analysis would behave if unprocessed with OPLS; the models are suboptimal in their discriminatory ability when OPLS is not applied as pre-processing method. The subplots from this figure show the following comparisons: (**A**) Controls (*n* = 110) vs. all cancers (*n* = 109). (**B**) Controls (*n* = 110) vs. endometrioid cancers (*n* = 57). (**C**) Controls (*n* = 110) vs. non-endometrioid cancers (*n* = 52). (**D**) Endometrioid cancers (*n* = 57) vs. non-endometrioid cancers (*n* = 52). (**E**) Controls (*n* = 110) vs. stage I cancers (*n* = 75). (**F**) Controls (*n* = 110) vs. stage IA grade 1 cancers (*n* = 22).

**Figure 4 cancers-14-05015-f004:**
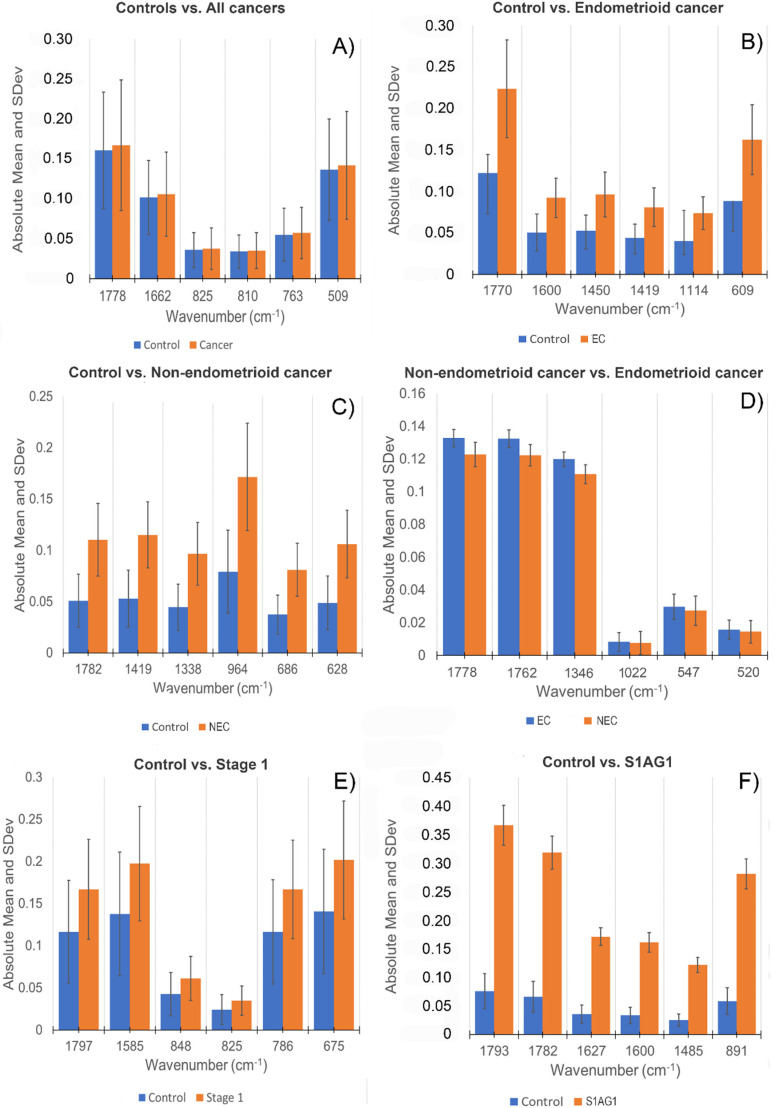
Bar plots from the spectral distribution and the mean absorbance, of the most discriminatory peaks found on (**A**) Controls vs. all cancers, (**B**) Controls vs. endometrioid cancers (EC), (**C**) Controls vs. non-endometrioid cancers (NEC), (**D**) Non-endometrioid cancers vs. Endometrioid cancers, (**E**) Controls vs. Stage 1 cancers, and (**F**) Control vs. Stage 1 grade 1A cancer classes.

**Figure 5 cancers-14-05015-f005:**
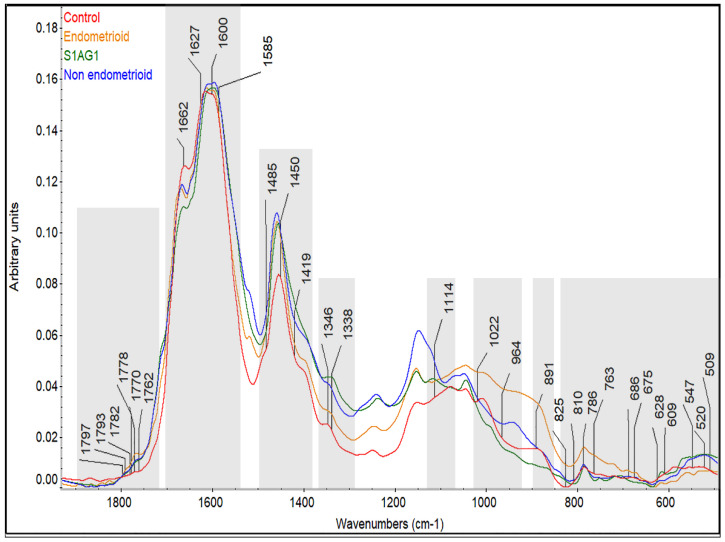
Average spectra of the studied groups. The figure depicts the fingerprint region (1800–500 cm^−1^) and the prospective spectral biomarks who are responsible for the discrimination between the analysed groups. The shaded regions indicate where the prospective biomarks are found, marking potential regions for endometrial cancer diagnostic.

**Table 1 cancers-14-05015-t001:** Patient demographics and clinical characteristics.

Patient Characteristics	Endometrial Cancers (*n* = 109)	Benign Controls (*n* = 110)
Age, years		
	67 (38–88)	56 (27–89)
<60 yrs	29/109 (26.6%)	60/110 (55%)
≥60 yrs	80/109 (73.4%)	50/110 (45%)
BMI (kg/m^2^)		
	31.9 (18.4–65.5)	27.8 (18.3–49.8)
<30 BMI	51/109 (46.8%)	79/110 (71.8%)
≥30 BMI	58/109 (53.2%)	31/110 (28.2%)
Type 2 Diabetes		
Yes, Diabetes	19/109 (17.4%)	9/110 (8.2%)
No, Diabetes	90/109 (82.6%)	101/110 (91.8%)

**Table 2 cancers-14-05015-t002:** Histopathological data for the study population.

Uterine Cancers		Benign Controls	
Endometrioid	*n* = 57	Ovarian cysts (non-endometriomas)	43
Grade1	32	Uterine fibroids and/or adenomyosis	27
Grade 2	17	Endometriosis (inc. endometriomas)	15
Grade 3	8	Endometrial polyps	4
Non-endometrioid	45	Uterine prolapse	3
Mucinous	1	Pelvic inflammatory disease	1
Clear cell	7	Endometrial hyperplasia	1
Serous	14	Cervical intraepithelial neoplasia (CIN)	1
Carcinosarcoma	14	Normal (no pathology identified)	15
Leiomyosarcoma	3		
Adenosarcoma	3		
Endometrial stromal sarcoma	2		
Sex cord tumour	1		
Mixed tumours	7		
Mixed endometrioid-serous	4		
Mixed endometrioid-clear cell	2		
Mixed endometrioid-serous-clear cell	1		

## Data Availability

Data are available from the corresponding author P.M.-H. upon reasonable request.

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
