# Peer review of "Infrared Spectroscopy of Urine for the Non-Invasive Detection of Endometrial Cancer"

_cancers, 2022, doi:10.3390/cancers14205015_

Round 1
Reviewer 1 Report
An interesting article.
In current form the article needs text and language editing.
Authors should also edit the reference section and pay attention to the way they were inserted into the manuscript. “Reference error” doesn’t allow to fluently read result’s section.
I suggest authors to improve the results explanation into the text. I found that is really difficult understand results by reading the graphs alone. A brief written comment could be really useful.
A question for authors; have you observed any type of correlation between spectra urinary sample and EC stage? It could be interesting to evaluate.
Author Response
Thank you for your review. Please refer to the attachment for my response.

Reviewer 2 Report
The article is devoted to a convenient and promising method for screening such a socially significant disease as endometrial cancer with means of infrared spectroscopy. The work definitely have a great importance and should be published.
Nonetheless, there are some questions.
1) Justify the choice of the methods of the analysis such as PCA and k-means as the clustering techniques, as well as used supervised analysis techniques.
2) Describe in results section, why do we see different heights for the same peaks in the control group on different graphs in Figure 4?

Author Response

(The authors gave the same response as above.)

Reviewer 3 Report
1. An overview about the study on the urine samples for cancer detection needs to be added in the introduction.
2. In the Method section, Line 94 “tumors were not classified to molecular profile” is confusing.
3. Comparing the unsupervised classification results by PCA, it highlights the effectiveness of supervised OPLS for binary classification, but the meaning of 4 clusters in Figure 1 is not very clear. The meaning of OPLS-PLS-DA for two clusters is also not clear.
4. Figure 4 compares the absorption values at several wavelengths in the infrared spectrum. Each panel needs to be numbered. The difference between (a) control and all cancers and (d) EC and NEC is little. It is suggested to screen out the spectral sites that can better reflect the difference.
5. Examples or typical spectra should be presented in the manuscript.
6. Table S2 is missing. The evidence that the level of nucleic acids in cancer is higher than control should be provided.
7. The analysis of the influence of high BMI, diabetes and age on the results has not been clearly demonstrated in the paper, and additional figures for comparison are needed to display this conclusion. Figure S9 does not address the problem. Moreover, the description of the overfit and underfit models in the figure caption of figure S9 may be confused, and the expression is inaccurate.
8. There is no description of Fig S10-S15 in the text.
9. Some editing errors need to be corrected, such as extra spaces on lines 92 and 96, as well as a number of incorrectly formatted references in the lines 159, 162, 182, 192, 199-209, 269, 273, etc.
Author Response

(The authors gave the same response as above.)

Round 2
Reviewer 3 Report
I am satisfied with the responses from the authors and their corrections regarding my comments. I have no further question in this submission.